# Needs of Sustainable Food Consumption in the Pandemic Era: First Results of Case Study

Laiza Andriolo da Rocha Ramos [1], Francesco Zecca [1,*] and Claudio Del Regno [2]

1    Department of Management, University of Rome La Sapienza, 00161 Roma, Italy
2    Department of Management & Innovation System, University of Salerno, 84100 Fisciano, Italy
*    Correspondence: francesco.zecca@uniroma1.it

**Abstract:** The current food system is directly associated with food insecurity, malnutrition, food waste, and environmental impacts. The international community has been working on sustainability, and the enhancement of sustainable food consumption is a fundamental step for identifying possible strategies to limit the negative consequences derived from the health emergency of the COVID-19 pandemic. This work aims to understand the food consumption patterns of the Sapienza University community. The methodology adopted for the research activity has been developed while taking into account the theoretical reflections and the tested methodologies acquired in relation to the subject matter. The survey was based on the acquisition of primary data obtained through the development and distribution of a questionnaire to a specific sample, the results of which have been translated into value terms in the form of indicators. The survey conducted had the purpose of carrying out a first evaluation able to provide some basic indications regarding the awareness within Sapienza of the relationship between sustainability and food. Based on the indications obtained at this stage, it is expected to give rise to additional and in-depth investigations aimed at providing a model of sustainable food consumption that can be replicated on a large scale.

**Keywords:** sustainability; food consumption; knowledge management

## 1. Introduction

The current food system is associated with food insecurity, malnutrition, and food waste or loss. World undernourishment has grown in 2020; between 720–811 million people had difficulty accessing adequate food, almost one in three people in the world [1]. Despite this, the global population will grow from 7.3 billion (2015) to 9.7 billion in 2050, and food production would have to double in magnitude in order to provide sufficient food [2]. Malnutrition is also a worry in 2020. In fact, there were millions of stunted, wasted, but also overweight children. Lastly, 30% of global food production is wasted or lost.

Furthermore, agriculture and food production play essential roles in environmental impacts [3]. While the former involves 40% of global land, the latter is responsible for 30% of global greenhouse gas (GHG) emissions and uses 70% of available freshwater [4]. The transformation of natural land into crop and livestock production is also classified as a threat to the extinction of species, soil degradation, and biodiversity loss. Additionally, marine life is affected since 60% of fish stocks are fished, and more than 30% are overfished [5]. It is impossible to feed 10 billion people in 2050 using current practices without destroying the planet [6].

Food and agricultural organizations (FAO) [7] classify GHG emissions from agriculture according to farm gate activities (livestock and crop activities) and land-use change; both correspond to 20% of all GHG emissions generated by humans. Another research showed that all stages of food systems contributed to 34 percent of anthropogenic global GHG emissions from 1900 to 2015 [8]. Meanwhile, Searchinger et al. [9] found that two-thirds of the worldwide agricultural land is destined for ruminant livestock, contributing to half of

agriculture's GHG emissions. Animal-based food will increase up to 68% by 2050, while ruminant meat demand will rise to 88% by the same year. Therefore, animal-based foods use more land, inputs, and GHG than plant-based foods. Xu et al. [10] conducted another study showing that the total activities of the food system corresponded to approximately 37 percent of total anthropogenic GHG emissions. Furthermore, animal-based products were responsible for 57% of this total; plant-based products were 29%, and 14% were attributed to other utilizations.

Based on the indicated information, it is incontrovertible that the current food system is driving the planet to its limits. Practices from production until the disposal of food need a shift urgently; otherwise, food insecurity and malnutrition will continue rising; global food waste will remain at 30 percent, and environmental degradation will still be a reality.

Therefore, it is necessary to transform the current food system into a more sustainable one, where overall economic, social, and environmental benefits are generated. Under the scope of the United Nations and with the main aim of defining a sustainable development project, the 2030 Agenda was adopted in 2015. The 17 sustainable development goals (SDGs) defined in the 2030 Agenda involve "citizens, private and public entities" [11]. In the 2030 Agenda, as a whole, particular attention was paid to improving a sustainable food system (SFS). Goal 12, with its 11 targets, 10 events, 13 publications, and 1190 events (by June 2022: from sdgs.un.org/goals), aims to promote new models of sustainable consumption and production by establishing a series of targets, such as achieving responsible management of natural resources and reducing food waste by 2030. Sustainable food consumption is present in goal 2 of the 2030 Agenda. This goal, in fact, promotes policies to eliminate world hunger, avoid food waste, and increase sustainable food production.

SFS is committed to delivering nutritious and safe food to all without compromising any economic, social, or environmental basis for future generations. It encompasses the sustainable food consumption and production [12]. To be sustainable, food consumption must be safe and healthy for consumers, generate decent income and livelihoods for the producers while not compromising the environment for future generations. The COVID-19 epidemic has had severe consequences on the entire food system. In fact, all the actors in the food systems have had shocking changes in their operations [13,14] (Dudek et al., 2022; Giudice et al., 2020).

Sustainable food consumption generates benefits for the consumer, the planet, and society [15]. The facilitation of introducing sustainable food consumption is based on two pillars: a demand side and a supply side [16]. From the supply side, having sustainable behavior means understanding the conducts appreciated by consumers and the ones that are not [17]. From the demand side, consumers influence the production patterns through their choices and lifestyles. Food consumption patterns are driven by disposable income, food prices, cultural and behavioral factors, education and available information, the infrastructure of the places they live in, and policy frameworks, such as local regulations and policies towards sustainable healthy diets and products [18]. Food consumption choices are subject to an assessment that causes reflection in terms of sustainability [19]. Scholars have argued that sustainable food consumer behavior implies understanding the differences between products in terms of sustainability of production processes and used resources [20–22]. Consumers' interest in sustainable food production and consumption increased the role sustainability plays in consumer choice of products [23].

The consumption of sustainable food has had a strong appreciation from Generation Z. The main reason is that Gen Z believes it is important to have a healthy lifestyle through the consumption of sustainable food, such as fresh fruit and vegetables on a daily basis [24].

Despite this, the literature [25–27] has perceived some obstacles faced by customers towards sustainable food consumption, such as the difficulty in accessing healthy food and information about sustainable policies, the gap between intention and action; the global resistance to adopting a diet characterized by the consumption of plant food and the poor variety of products in supermarkets, among others.

The United Nations Food and Agriculture Organization claims that much of the food produced in the world is not consumed and becomes waste food [28]. Obviously, the lack of food has serious economic, environmental, and social consequences. Considering the social aspect, already in 2013, approximately 800 million people suffered from underconsumption of food and difficulties in obtaining drinking water [29,30]; on the other hand, 1–1.5 billion people are overweight and 300–500 million of them are obese [31]. This investment can impact farmers, decreasing their incomes and increasing consumer spending [32]. Consumption of food is also considered an "essential and unavoidable part of everyday life, and it is an area where personal consumption is closely related to environmental sustainability" [33,34]. Scholars have paid attention to the relationship between sustainability and sustainable consumption behaviors [35–38] and the segmentation of sustainable food [33,39–46].

There is interesting recent research published around this relationship (sustainability and food consumption) during the period of a health emergency [47–50]. Rodriguez-Basteiro and Valencia-Zapata [51] contribute to the understanding of how university students perceive the risks of COVID-19 in their food consumption.

While the relationship between sustainability and sustainable consumption behaviors has been investigated, and also this relationship in the context of risks refers to university students, the aspect of consumption of sustainable food by university students during the period of COVID-19 has not really been investigated. From the extant literature, no previous attention has been paid to the consumption of sustainable food by the members of Generation Z (or university students). The last similar journal paper detected in the literature was published in 2019, and it aimed to segment generation Z by using sustainable food consumption behavior, but COVID-19 was not part of the research. Therefore, from the literature, it is detected that there is a research gap on the topic addressed in this paper.

The so-called Gen Z, or the new generation born in the mid-1990s, count for approximately 2 billion individuals worldwide [52], so the consumption behaviors on sustainable food during COVID-19 for Gen Z in the university context is intended as a relevant research area through which new knowledge can be obtained. As exploratory research, this paper aims to analyze food consumption in the context of Italian public universities. The work explores the question of Gen Z university students studying in Italian public universities who have adopted SFC behavior. It is important to verify how COVID-19 has influenced these behaviors.

All this considered, the following research questions have been developed:

(I)　Is there a gap between intention and action regarding sustainable food consumption during COVID-19?
(II)　Does the fish diet have its own specific and different sustainability?
(III)　Do people who usually follow a plant-based diet tend to have more sustainable habits than those who follow an animal-based diet during COVID-19?
(IV)　Do university students belonging to Generation Z have sustainable food consumption behavior during COVID-19?

The main goal of this paper is to identify food consumption patterns in the university community, through an analysis of Sapienza University, revealing if their food habits are in line with sustainability. Second, the research aims at investigating consumer solutions to bettering Sapienza's sustainable food system. The empirical verification was conducted based on a questionnaire.

It is essential to emphasize that the sample used for the survey was composed of highly educated people living in a high-income country and potentially interested in sustainability. In fact, according to Antonelli et al. [53], high-income countries tend to be more sustainable than developing countries and less developed countries (LDCs).

In addition, highly educated people tend to be more sustainable than less educated people. According to Žaleniene and Pereira's [54] research, countries with high rates of university enrollment tend to present good economic, social, and environmental perfor-

mance. The conclusions have provided considerations in order to introduce sustainability as a significant element of food consumption in Sapienza's community.

## 2. Theoretical Background

### 2.1. Food Consumption of Gen Z during the CVOID-19 Pandemic

Gen Z represents an important generational cohort in society. These are individuals who are between 17 and 23 years of age, and most of them live alone and, therefore, have to make independent food and consumption decisions [55]. In particular, these are university students who make different consumption choices based on their lifestyle habits. Gen Zs who live away from home consume wrong and unsustainable food [56], while those who live at home have food choices that are more in line with the principles of sustainability [57]. This choice is also determined by the family conditions of these consumers because they follow the lifestyle of their parents [58]. The level of education also strongly influences the food consumption behaviors of Generation Z. In fact, university students belonging to this generational cohort prefer sustainable food consumption and adopt a sustainable dietary lifestyle [59]. Furthermore, customer knowledge management allows consumers to dynamically capture, create, and integrate knowledge on certain goods, influencing their consumption choices [60].

The COVID-19 epidemic in 2020 impacted consumers' food consumption behaviors [61]. Lately, there has been a growing trend in the consumption of sustainable food. In fact, 2020 showed that during the period of a health emergency, the consumption of fruit and vegetables increased, which is necessary to improve immunity, resulting in the adoption of a healthier diet, and a sustainable lifestyle [24].

### 2.2. Food Consumption and Sustainability

Food consumption referred to as sustainability (SFC) is based on the integrated implementation of sustainable food consumption and production patterns, without having consequences on ecosystems [62].

Many studies referring to the consumption of sustainable food have focused on specific and singular foods linked to sustainability, such as organic foods [63,64]. Indeed, organic food is an important example of a sustainable food product [34]. According to Quoquab and Mohammad [65], the consumption of sustainable food guarantees three important elements: quality of life, respect for the external environment, and satisfying the needs of future generations.

The literature suggests that there are many factors influencing sustainable food consumption behaviors, such as positive attitudes towards sustainable food, social and personal norms, and social context, especially in the family [66].

The family context influenced the behaviors of young consumers [67–70]. At the same time, consumer behavior is influenced by the behavior of companies. Especially during the COVID-19 pandemic, companies had to influence the minds of consumers to get their shopping habits to resume [71]. Marketing strategies promoted by companies are also strictly linked to consumer behavior [72].

Social media also has a strong influence on Gen Z consumption behaviors. In fact, much research has been conducted in the context of the use of social media on adolescents and its impact on employee well-being, including in the terms of food consumption [73].

According to Annunziata [43], there is a strong influence on the structure of contemporary Italian families on the sustainability of food models, especially regarding the consumption of organic food.

Sustainable consumption, therefore, has become an integrated aspect of sustainable development [74]. The changes toward more sustainable consumption patterns are useful for solving numerous problems related to sustainability [75].

*2.3. Relationship between Food Consumption by Gen Z and Sustainability during COVID-19 Pandemic*

Guaranteeing the consumption of sustainable food is one of the objectives of the 2030 Agenda (objective no. 12). For this reason, political leaders have focused on SDG 12 to increase efficient food production policies in order to minimize consumption and waste. All this would lead to improving the sustainability of food supply chains [76].

Food production and consumption generate impacts on the environment, such as greenhouse gas emissions, water pollution, and biodiversity loss [31]. Consequently, the choices related to food consumption are of great importance in relation to the environment [55], causing serious consequences for future generations. In recent years, numerous studies have found that organic production and consumption are ways to solve the problem of food waste [77]. According to Perez [78], organic food reduces the negative consequences of agriculture on the environment. For university students belonging to Generation Z, research realized in 2007 revealed a growing propensity for young consumers to consume healthy food to improve the quality and longevity of life and prevent emergencies and diseases, such as obesity [79].

Despite this, the literature has brought out different behaviors among university students. In particular, some research suggests that students living away from home have lower sustainable consumption behaviors than those living at home [80]. Beretzky and Jámbor [81] argue that Gen Z college students prefer the consumption of sustainable and healthy foods, with the choice of consuming fast food occasionally. Ahamad et al. [82] claim that there is significant knowledge on sustainable consumption among students belonging to Gen Z in universities, even if their sustainable consumption practice is moderate. The improvement of sustainable food consumption has increased through the education of students [83]. According to Muresan [84], sustainable eating behavior during the COVID-19 pandemic is influenced by education.

Finally, according to this literature, we can agree that the behaviors on food consumption are influenced, among other things, by an attention to health implications also among the Gen Z demographics.

## 3. Methods and Data

From our exploratory research, this paper uses a qualitative approach. Case study research [85] has been adopted as a methodology approach in line with the aim of this paper. Within the case study research, a single case study has been used as a research strategy, and the Sapienza University in Rome (Italy) has been selected as the object of our investigation. This choice derives from a number of factors. Firstly, this university is number one in Italy for the number of students enrolled [86]. This was very important because it allowed access to a larger sample than other European and Italian public universities. Furthermore, another fundamental point is the ease of obtaining data and information useful for research.

The methodology adopted has been developed on-going with the research process, considering the theoretical reflections and the testing methodologies [35,40,84] acquired in relation to the main aim of the research. On the basis of the relationship among the elements of the theoretical background and in line with the aim of the paper, an online semi-structured survey has been developed based on the acquisition of primary data with a specific target. This approach presents the methodology of case study research by assuming "a survey within a case study" [87].

The sample has been created on the basis of the availability to voluntarily participate and complete the questionnaire (availability sample). Overall, the randomly formed sample turned out to be created with 780 people united by a work or study-related bond with the university. This last one has represented the only condition at the base of the sampling's specificity.

The questionnaire has been designed using previous studies in order to obtain clear and schematic information related to the type of research that was carried out. The research activity was in fact exploratory, and it was based on the collection of useful data to be used

for subsequent investigation within the same reference framework. The questionnaire's validity and reliability analyses were conducted.

The design of the questionnaire in each question would consider responses that could be measured on a two- or four-point Likert scale by reason of the previously assumed depth level of the investigation. This type of scale has in fact been used because it is able to provide immediately applicable results. Aside from the ordinal scale questions, the questionnaire concluded with an open-ended question that could collect suggestions from respondents on how to possibly improve the level of sustainability of food consumption within the university. According to the literature [88], word frequency accounting has been the method used to analyze the sample's opinion on their sustainability level. The inclusion of this last question had the objective of translating the specific research question concerning the ability of Sapienza University to create a sustainable supply chain.

In order to synthesize the frequency distribution of responses and turn the information provided into unique values, the indicators used for the composition of the sustainable consumption index (SCI), environmental consciousness index (ECI), waste segregation and recycling index (WSRI), and healthy nutrition index (HNI), have been considered. The variables included in each index were selected based on the literature. The construction of the indexes has been inspired by the Barilla Foundation sustainability index [89].

For each of the three indicators used for the composition of the sustainable consumption index, the characteristic values are able to adequately represent the sustainability trends detected and compare them with the working research questions that have been identified. Phyton was the software used for transforming the answers to the questionnaire into indicators with values from 0 to 1. The use of indicators had the objective of summarizing the knowledge acquired from the cognitive investigation and checking how the answers to the different questions were intertwined.

## 4. Results

The sample analysis started with general questions regarding sustainability: "how is your knowledge regarding the topic sustainability?" and "do you think your food consumption is sustainable?". Figure 1 comparing the answers.

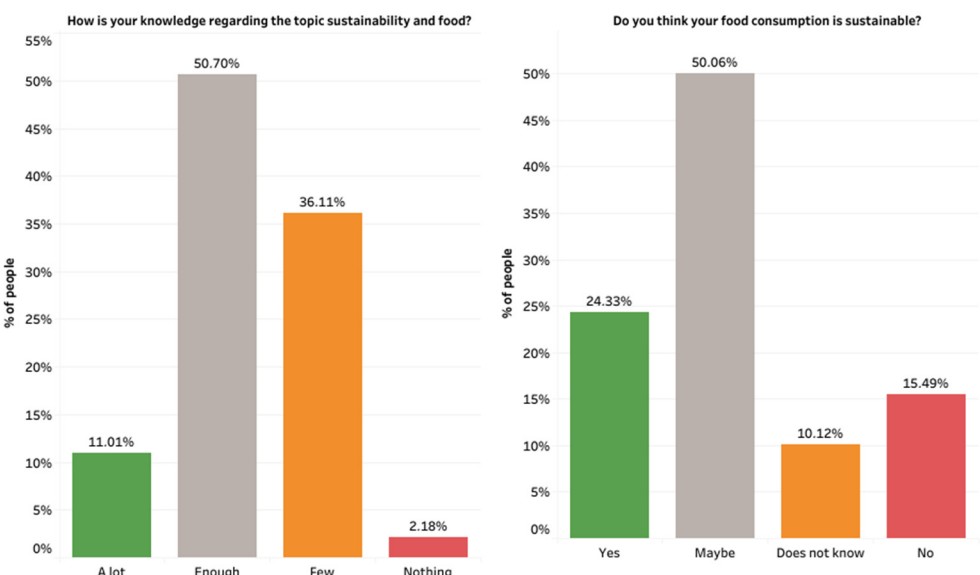

**Figure 1.** Sustainability knowledge versus sustainable food consumption.

The first graph shows that half of the sample had enough knowledge about sustainability, whereas only 11 percent were fully aware. The most interesting point is that 38 percent claim to have little or no knowledge about the topic.

On the other hand, the second graph demonstrates that just 24 percent of people believed their food consumption was sustainable. In comparison, more than half stated "maybe," 10 percent answered, "does not know," and 15 percent claimed to not have sustainable habits.

### 4.1. Intention versus Behavior

This section analyzed only consumers who believed their food consumption was sustainable, comparing their answers to other related sustainable actions. In summary, two gaps were found when they were asked about how frequently they purchased products in sustainable stores and when asked if they bought fish from sustainable retailers. In the Figure 2, it is possible to see them.

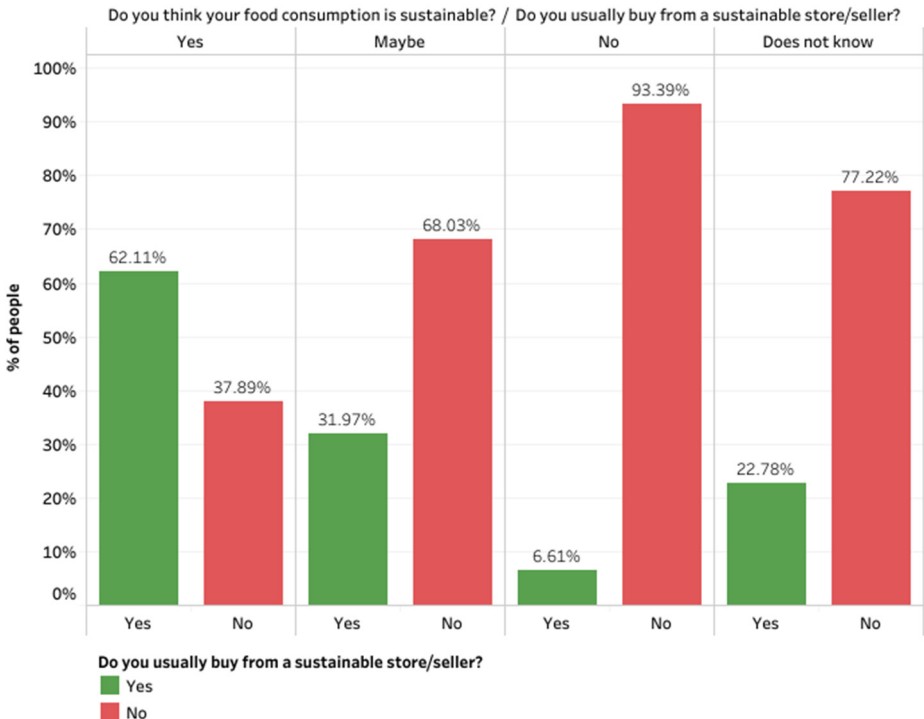

**Figure 2.** Sustainable food consumption versus sustainable store/seller.

The most interesting fact is that 68 percent of those who may have had sustainable food consumption habits did not buy from sustainable retailers; it was more than double compared to those that usually bought from sustainable sellers.

At the same time, not all people who affirmed their consumption was sustainable, purchased from sustainable retailers. In general, people did not give so much attention to sustainable stores/sellers; a total of 65.56 percent answered no, whereas 34.44 percent answered yes. The Figure 3 comparing the answers.

In total, 25.2 percent of people affirmed they never bought fish from a sustainable retailer; precisely, 41.7 percent answered "rarely"; exactly, 28.2 percent answered "sometimes"; and only 4.9 percent answered "yes". The most surprising discovery was that, of those who believed their food consumption was sustainable, only 11.6 percent actually bought fish from sustainable stores and 33.7 percent sometimes purchased. On the other hand, 28.4 percent rarely purchased fish from sustainable stores, and 26.3 percent never purchased.

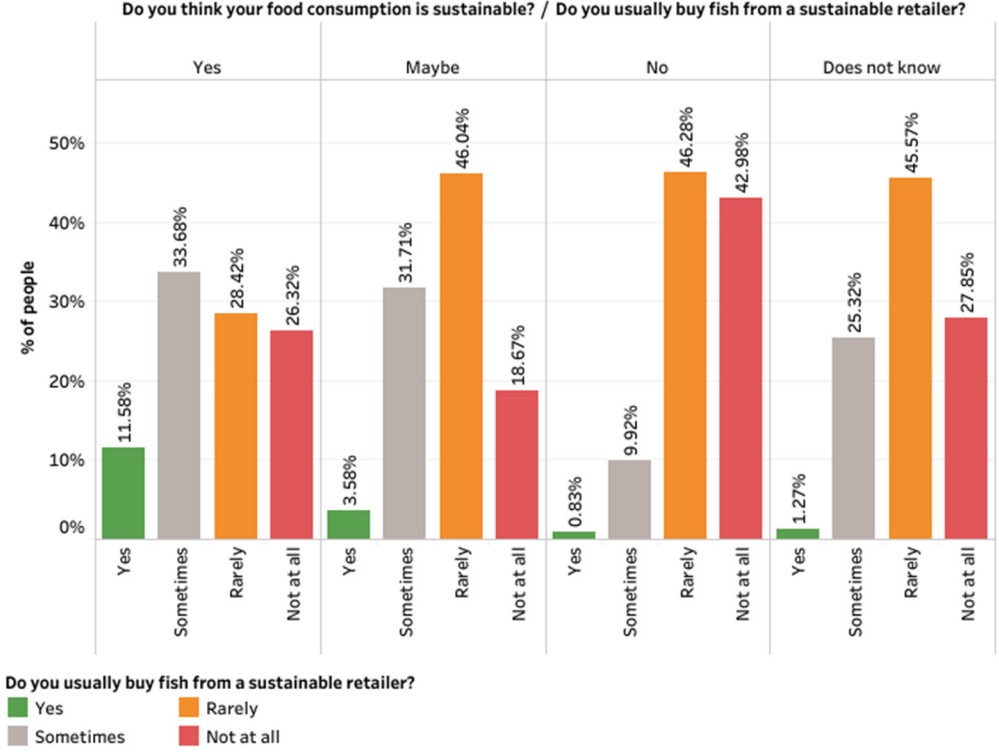

**Figure 3.** Sustainable food consumption versus sustainable fish retailer.

Of those who may have had a sustainable consumption, just 3.6 percent bought fish from sustainable retailers, and 31.7 percent sometimes purchased, whereas 46 percent stated "rarely", and 18.7 percent answered, "not at all." The results demonstrated the lack of awareness of sustainable fisheries and their importance in the ecosystem and the SDG achievement.

*4.2. Plant-Based Diet versus Animal-Based Diet*

The objective of this section was to understand if the sample tended to choose plant-based or animal-based products and, at the same time, analyze if people who followed a plant-based diet had more sustainable actions. In this survey, 62.23 percent of people chose plant-based food, while 37.77 percent usually chose animal-based products. The Figure 4 compares the different types of diets and sustainable food consumption adopted by consumers.

Firstly, it is essential to consider that almost two-thirds of the sample chose plant-based products.

After analyzing this type of diet, 31 percent had sustainable food consumption; nearly 50 percent may have had it, and 19 did not have it or did not know about it. Regarding the animal-based diet, only 13.5 percent had a sustainable consumption; 51 percent may have had it, and 35.5 percent did not have it or did not know it. In summary, people who chose plant-based products were more sustainable than those who chose animal-based products.

The following figures (Figures 5–7) have the same purpose, analyzing each type of diet compared to the following topics: purchase food from sustainable sellers, attention given to eco/sustainable products, and the purchase of seasonal products.

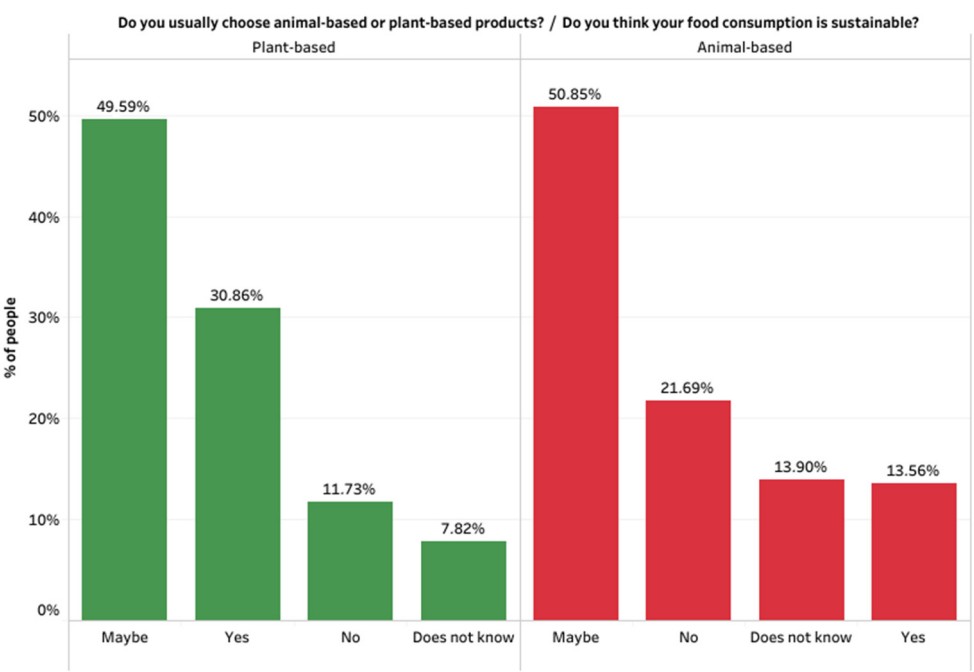

**Figure 4.** Sustainable food consumption versus type of diet.

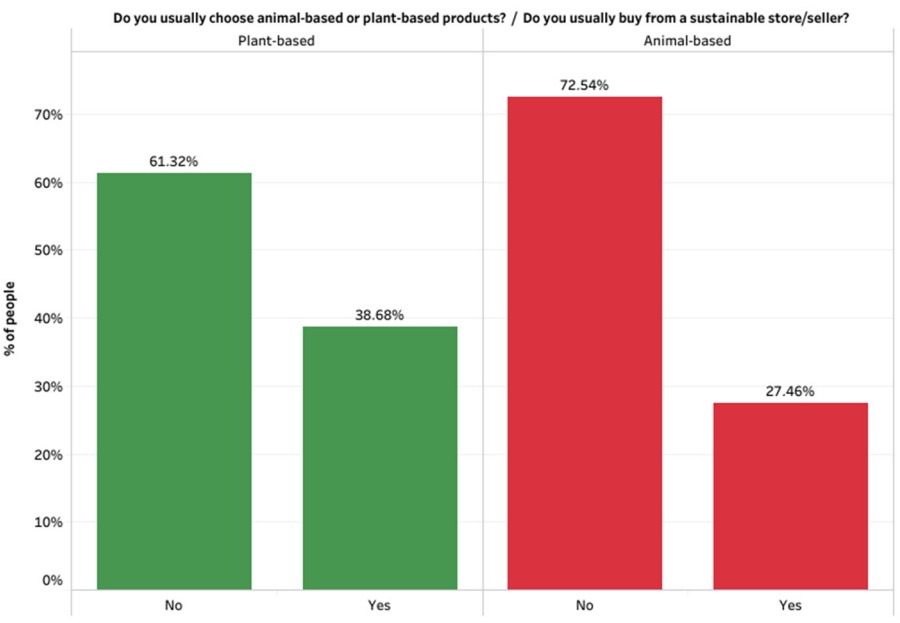

**Figure 5.** Sustainable diet versus sustainable store/seller.

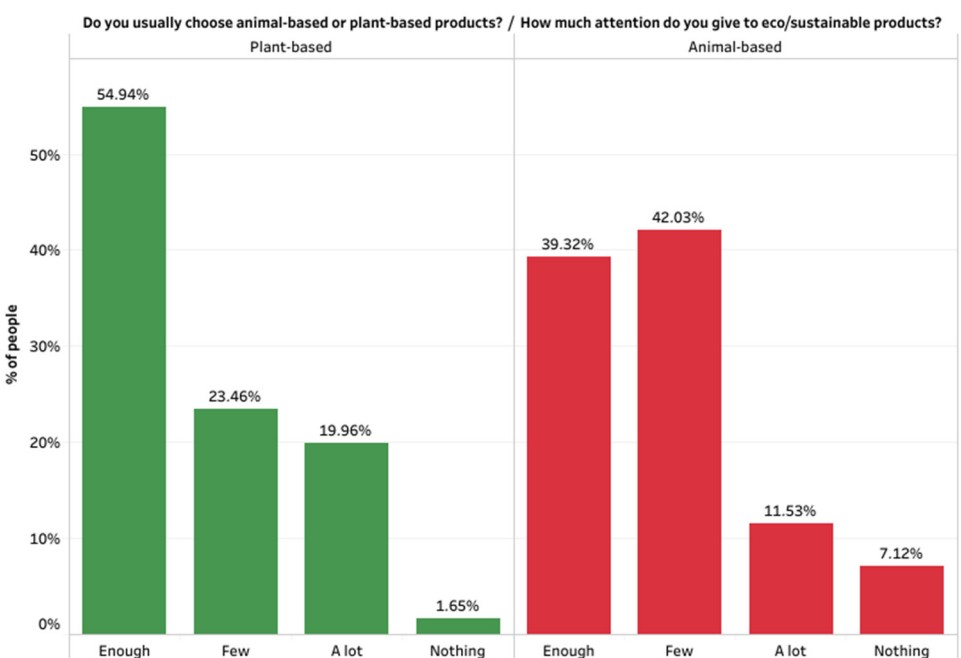

**Figure 6.** Type of diet versus attention to eco/sustainable products.

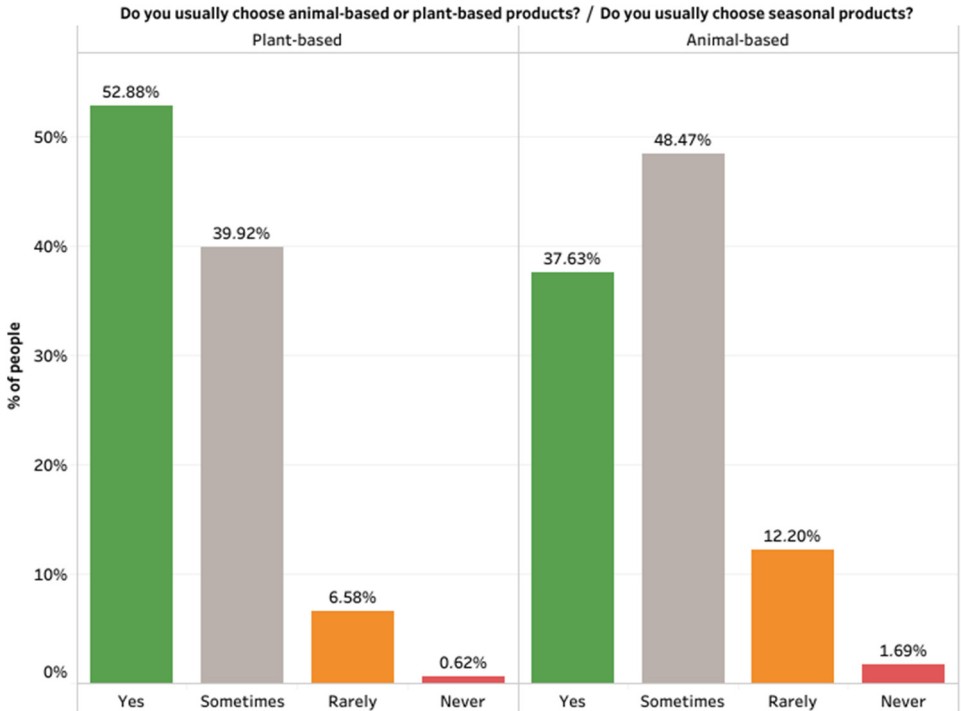

**Figure 7.** Type of diets versus seasonal products.

All the figures above verified the previous hypothesis: people who followed a plant-based diet tended to encounter more sustainable habits than those who preferred animal-based products. The only topic that did not show any difference was the one that referred to separate waste collection: a total of 83 percent used separate waste collection, and 17 percent did not use it for both types of diets.

### 4.3. Food Supply

In this section, it was possible to identify the most popular places chosen for food consumption in the Sapienza University community. Moreover, the objective was to visualize places inside the university, such as bars, vending machines, and the university canteen, and their food supply patterns. Figure 8 shows the usual general places of consumption.

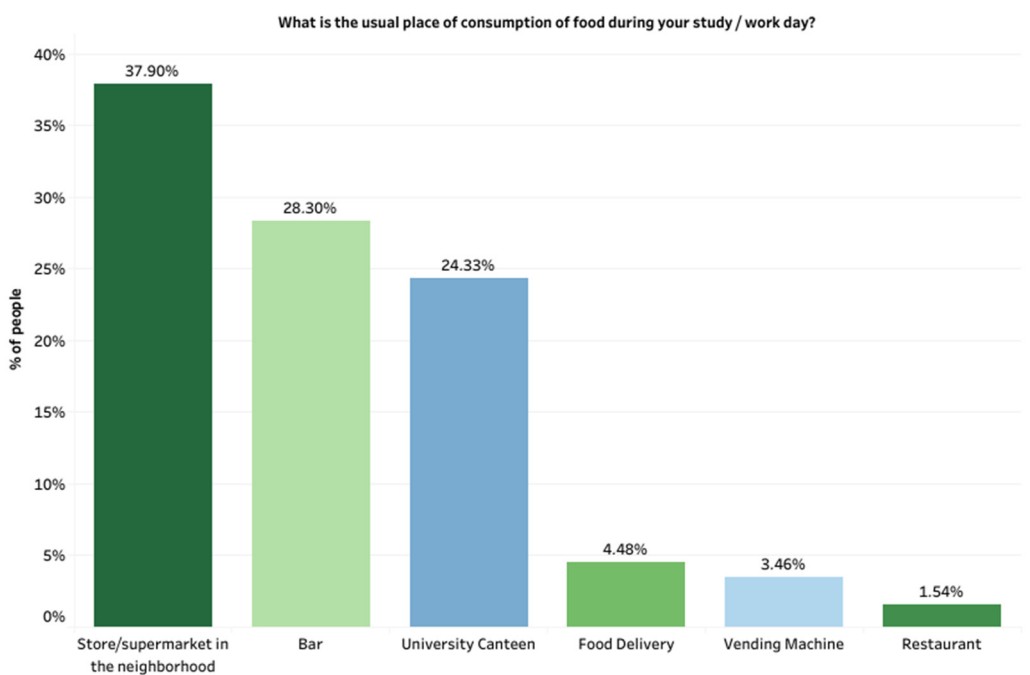

**Figure 8.** Usual places of consumption.

To understand the food supply of Sapienza University, the following figures (Figures 9 and 10) were analyzed, focusing on the places inside the university: canteen, vending machines, and bar, which were called group 1. Group 2 covered people who usually ate outside of the university: food delivery, store/supermarket, and restaurants.

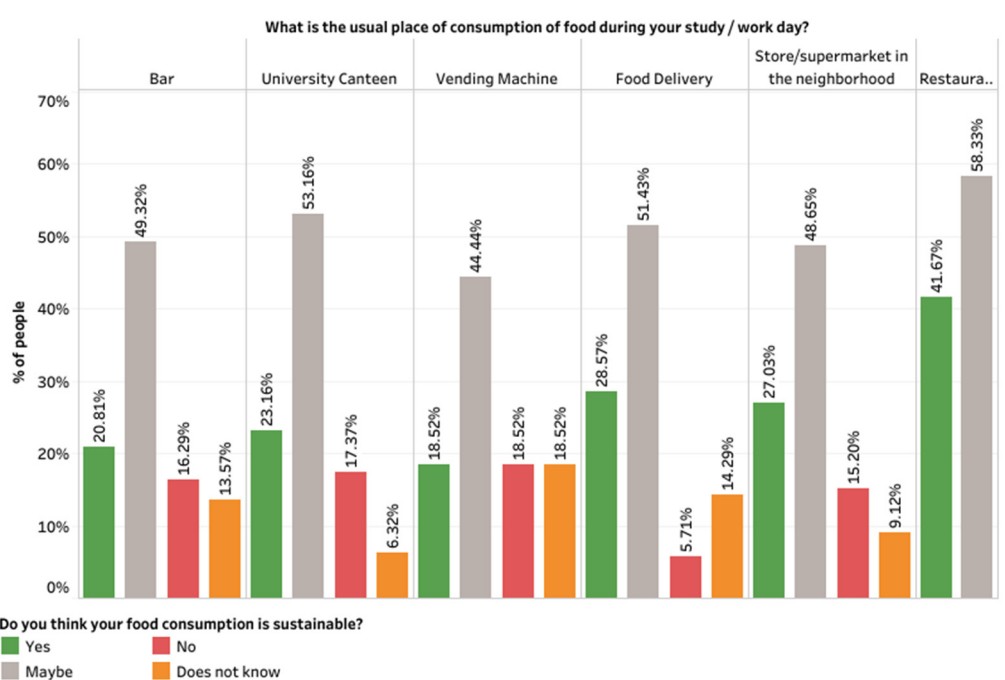

**Figure 9.** Places of consumption versus sustainable food consumption.

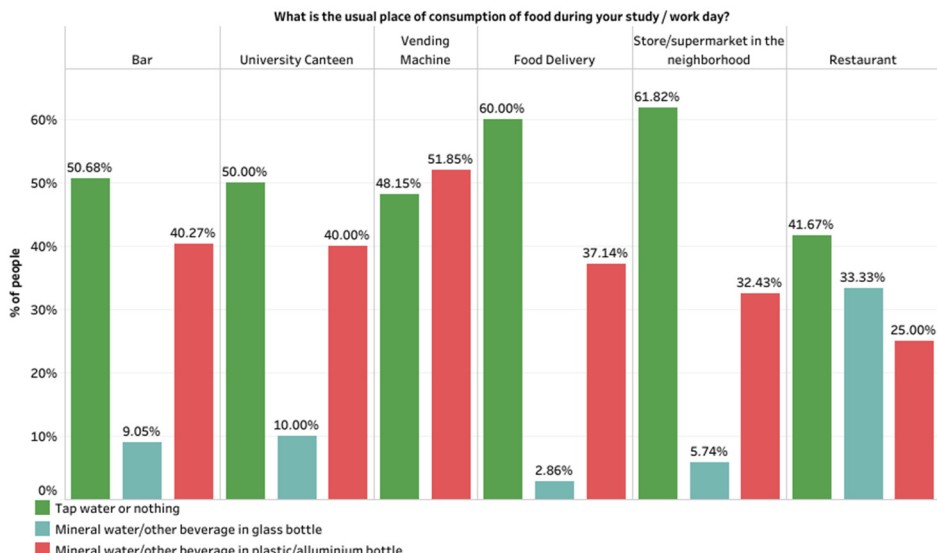

**Figure 10.** Type of beverage versus usual place of consumption.

While Figure 9 shows that fewer people from Group 1 considered their food habits sustainable when compared to Group 2, Graph 10 demonstrated that approximately 40% of people who ate in the bar and university canteen still purchased beverages in plastic bottles and 52% bought food from vending machines. Additionally, topics about healthy nutrition raised concerns about the Sapienza food supply. When people were asked about their consumption of whole grains, the composition was as follows: Approximately 21% from Group 1 answered "a lot" compared to 34% from Group 2; and 41% from Group 1 answered "few" or "nothing" compared to 23% from Group 2. When the question was "How often do you eat fruits and vegetables?" approximately 29% from Group 1 answered "a lot," compared to 48% from Group 2; and 25% from Group 1 answered "few" or "nothing" compared to 11% from Group 2.

### 4.4. Sustainable Consumption Index

In this section, the SCI was developed. The index converted all answers into values; each answer corresponded to a score from 0–1. In particular, 0 referred to the least sustainable, and 1 referred to the most sustainable. The SCI considered three dimensions of the survey: ECI, WSRI, and HNI.

### 4.4.1. Environmental Consciousness Index

This dimension refers to the questions covering environmentally sustainable actions. The set of six questions was the following: "Do you usually choose animal-based products or plant-based products?", "Do you tend to reduce red meat consumption?", "Do you usually buy fish from a sustainable retailer?", "Do you usually choose seasonal products?", "How much attention do you give to eco/sustainable products?", "Do you usually buy from a sustainable store/seller?". All the answers were transformed into values. For instance, regarding the first question, "animal-based" is equal to 0, and "plant-based" is equal to 1. All the other answers followed the same patterns: "nothing/not at all" is equal to 0.25; "few/rarely" is equal to 0.5; "enough/sometimes" is equal to 0.75; and "a lot/yes" is equal to 6 (highest score/most sustainable). Figure 11 was created.

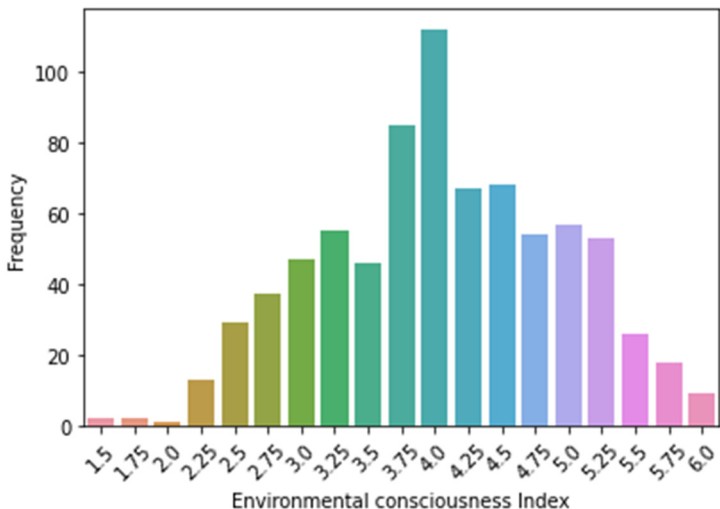

**Figure 11.** Environmental Consciousness Index (ECI).

The mean of the ECI was 4.06; the mode and median were equal to 4. No one received 1.25, the minimum value, while very few people took the maximum value. After that, the normalization of values was performed, creating a set of values from 0 to 1—transforming the mean into 0.57.

4.4.2. Waste Segregation and Recycling Index

In this dimension, the following questions were analyzed: "What kind of beverage do you drink during your meal?" and "How do you discard your waste after eating?". The first issue referred to the utilization of plastic, and the second one mentioned the discard of food using separate waste collectors. Concerning the first question, the answer "Tap Water or nothing" was equal to 1. In contrast, "Mineral water/other beverages in glass bottles" was equal to 0.5, and "Mineral water/other beverages in plastic bottles" was equal to 0. Regarding the second question, the answers "In the place of study/work using separate waste collection" and "In places other than the place of study/work using separate waste collection" were equal to 1; and the rest were equal to 0. In this dimension, each person may score from 0 (lowest score/least sustainable) to 2 (highest score/most sustainable). Figure 12 was created.

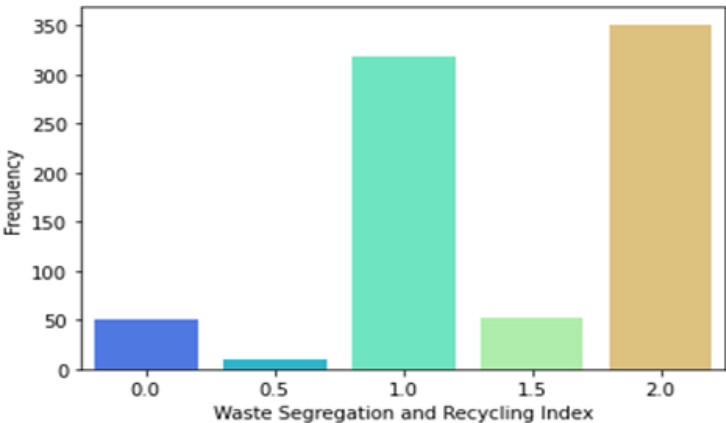

**Figure 12.** Wast. Segregation and Recycling Index (WSRI).

The mean of the WSRI was 1.41; the mode was 2, and the median was 1.5. Most people received the highest score; the sample had a good performance when considering separate waste collection. After that, the normalization of values was performed, creating a set of values from 0 to 1, transforming the mean into 0.71.

### 4.4.3. Healthy Nutrition Index

In this dimension, a set of five questions was analyzed, aiming to understand if the sample had healthy food habits. The questions were the following: "How many fruits and vegetables do you consume?", "How often do you choose fresh products/ingredients?", "How often do you eat whole grains?", "How often do you eat legumes?", and "Do you drink sugary beverages?" All the answers were transformed into the following values: "a lot" was equal to 1; "enough" was equal to 0.75; "few" was equal to 0.25; and "nothing" was equal to 0.25; while the answer to the last question had the opposite value. In this dimension, each person may score from 1.25 (lowest score/least sustainable) to 5 (highest score/most sustainable). Figure 13 was created.

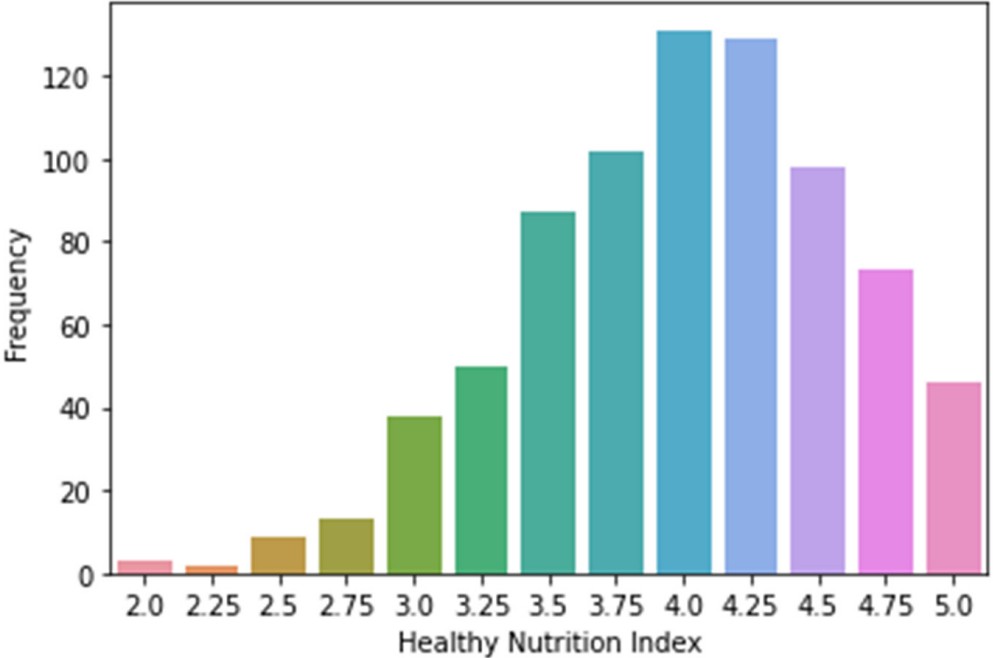

**Figure 13.** Healthy Nutrition Index (HNI).

The HNI had 3.5 as the mean, mode, and median. No one received the least healthy score, and some people received the highest one. After that, the normalization of values was performed, creating a set of values from 0 to 1, transforming the mean into 0.6.

### 4.4.4. Final Result

SCI included the sum of all three dimensions already discussed: environmental consciousness, waste segregation and recycling, and healthy nutrition.

Each person represents a value from 0 to 3. In particular, 0 referred to the least sustainable, and 3 referred to the most sustainable. The values were normalized, and the index was finalized within a score from 0 to 1, generating the Figure 14.

The SCI of Sapienza University had its mean equal to 0.67, the mode equal to 0.76, and the median equal to 0.68.

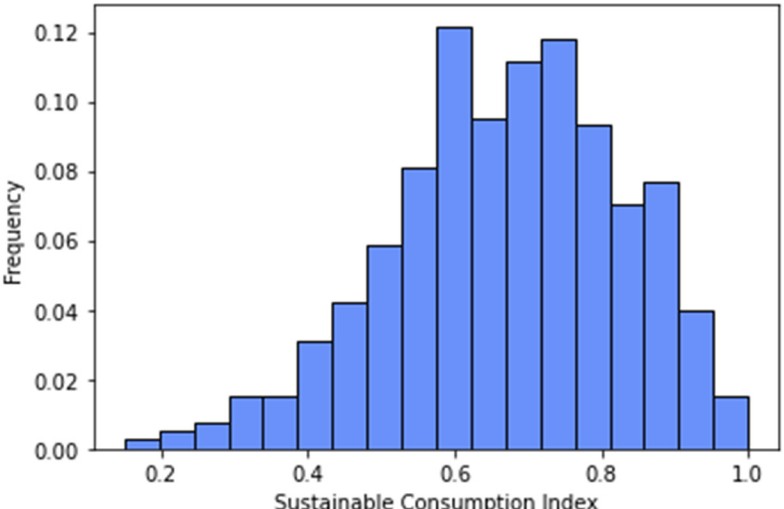

**Figure 14.** Sustainable Consumption Index (SCI).

4.4.5. Open-End Answers: Solutions for Sustainable Food Consumption

The last topic of the survey was an open-ended question, where all participants were free to write their own suggestions and concerns about sustainable food consumption. Through the responses to "what solutions do you suggest making the food consumption model more sustainable?", it was possible to identify the most repeated words, understand how people see the problem, and the solutions proposed. All points, commas, prepositions, spaces, and irrelevant words were removed in this method. For a better analysis, similar words were grouped, and only those that were fundamental to the study remained. Figure 15 shows the most repeated words in the open-ended question of the questionnaire.

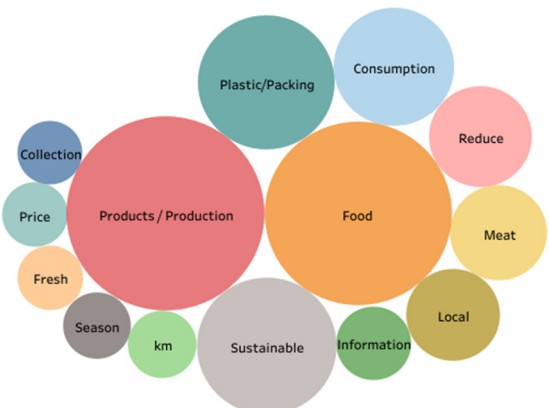

**Figure 15.** Keywords suggested for sustainable food consumption.

According to the research, the above words were repeated with the following frequency: Products/production (308 times); Food (275 times); Sustainable (153 times); Plastic/packing (147 times); Consumption (114 times); Reduce (83 times); Meat (74 times); Local (69 times); Information (44 times); Km (37 times); Season (36 times); Fresh (35 times); Price (34 times); and Collection (33 times).

## 5. Discussion

The results demonstrated that 50 percent of the sample had enough knowledge regarding sustainable food consumption, while 11 percent affirmed, they knew a lot about it. Even though it seemed a favorable outcome, when they were asked if their food consumption was sustainable, only 24 percent affirmed it was and 50 percent stated "maybe". Considering the answers come from highly educated people in a high-income country, more involvement

with sustainability was expected. This result, therefore, appears to confirm what is indicated by the literature previously mentioned regarding the lack of education on sustainability, even if the same deficiency has been found in a segment of the population that other studies [90] have indicated as more attentive to the theme. However, some percent of the sample does not appear to be far from the SDG 4.7 of the 2030 Agenda concerning the acquisition of all the skills and knowledge useful for leading sustainable lifestyles.

The first research question analyzed in the survey was a possible gap between intention and action, as summarized in Figure 3.

The analysis of the results confirms the research question of a gap between intention and action for approximately 1/3 of the sample. The results obtained seem to be in contradiction with what has been highlighted regarding the general theme of the level of knowledge of sustainable food consumption. It would have been reasonable to expect a higher percentage of the sample regarding the intention-to-action gap. Most likely, the intentions of the sample are independent of the level of knowledge on the subject and highlight a priori trust in retailers of sustainable products.

Another gap was observed when analyzing the figure for sustainable food consumption versus sustainable fish retailers. The most interesting point was that one-quarter of the total participants never bought fish from sustainable stores and 41 percent rarely purchased fish from sustainable stores. The results obtained do not seem to confirm the second research question, putting forward, in the literature, a specific and different sustainability question on fish diets [91]. The sample interviews had in fact highlighted that 2/3 of the participants lacked knowledge of the possible link fish have on sustainability.

The third research question discussed the tendency of consumers that had a plant-based diet to be more concerned about sustainability or more sustainable compared to those who followed an animal-based diet. Overall, the research question was confirmed in this survey. The only topic that demonstrated no difference regarding the two diets was the one related to separate waste collection: a total of 83 percent of the participants—from both groups—separated their waste. These obtained results were in contradiction with the consideration that sustainability remains marginal in daily food choices [66,92].

The last research question covered students belonging to Gen Z, who have sustainable food consumption behavior, was satisfied.

The outcome demonstrated that people who usually ate in the bar, university canteen, or vending machines had lower adherence to plant-based diets compared to other places; at the same time, they tended to eat fewer legumes and whole grains. On the other hand, they gave the same attention to eco/sustainable products and had the same fruit and vegetable consumption patterns compared to the other places. The most concerning point was the use of plastic/aluminum bottles inside the university, showing that people who usually consumed in the university canteen, bar, and vending machines tended to consume more plastic than in the places outside of the university. These results highlight the gap between the level of sustainability of demand and the level of sustainability of the food offered in internal supply chains and provide evidence of the crucial role of sustainable food supply chain management [93].

Therefore, it seems the university should incentivize the sales of plant-based products inside the university, either by increasing the variety of these items or lowering their prices. One proposal could be partnerships between the university and local ecological fairs, enhancing the work of smallholder farmers and the quality of their fruits, vegetables, and whole grains—resulting in fairer prices and higher demands.

The fourth result discussed was SCI, which considered three important indexes for estimating the sustainability value: ECI, WSRI, and HNI.

The final score of ECI was 0.57. It demonstrated that people were aware of the environmental impact caused by food consumption; however, they did not give much importance to the place of purchasing. Most people buy their food in supermarkets, the most popular place for purchasing food, and they will not change this habit. The use of supermarkets is linked to the greater economic sustainability of food consumed.

The WSRI final score was 0.71 indicating the best performance of the used indicators.

The final score of HNI was 0.6. This result confirmed that the Sapienza community was sufficiently aware of the nutritional characteristics of the food they consume.

The value of the SCI result was 0.67. It was not an unsustainable value since it was above the average; however, it could be much better, considering the limitations of the sample—highly educated people potentially interested in sustainability residing in a high-income country.

Finally, the last question discussed the most frequent words found in the suggestions. "Products/production" was the most repeated word (308 times) and was usually accompanied by the word "Sustainable" (153 times). People often suggested purchasing sustainable products; however, most of them did not buy from sustainable sellers, as analyzed before. Those who cited the word "Products/production" often mentioned the necessity of more places inside the university selling eco/sustainable products. At the same time, they emphasized that these items must contain better identification on their labels, calling attention to the public.

"Food" appeared 275 times, the second most repeated word, exposing the awareness of the sample about how impactful their food habits are and how fundamental it is to choose the right products for achieving sustainability. This word was usually close to others, such as "Local" (69 times), "Season" (36 times), and "Fresh" (35 times). Previously, the sample showed good performance when asked if they chose seasonal/local products, corroborating their perception of the environmental benefits of these types of food and their effective action when buying them. The freshness of the food was also considered in the creation of the HNI, since 95.5 percent of people tended to choose a lot/enough fresh ingredients. Therefore, some people emphasized the need to have fresh products for sale inside the university, suggesting the lack of it, another improvement Sapienza needs to consider.

Another word that accompanied "food" was "Km" (37 times), or "km 0". These products are commercialized near the place of production, so they do not travel kilometers for their consumption. They are usually non-industrialized products, enhancing healthy nutrition due to their freshness and quality. Additionally, they do not create environmental consequences, by not requiring high levels of energy for storage or polluting via transportation. Moreover, it is interesting that people brought this concept to the survey, even though it was not mentioned in the questionnaire, showing their knowledge of these products and their advantages.

When observing the word "food," many people cited the reduction in food waste to achieve sustainability. As already discussed, 1/3 of the global food production is lost, and the sample seems to be aware of that. Furthermore, they highlighted the necessity of paying attention to the amount of food purchased since a great part of food waste happens at the household level. The word "collection" appeared 33 times, and some suggestions raised the importance of separating waste correctly. Separate waste collection provided surprisingly one of the best outcomes, since 82 percent of people practised it, regardless of the place of discarding.

"Consumption" (114 times) was a very frequent word as well, and "Plastic/packing" (147 times) was frequent on the side of this word, demonstrating that people were aware of the damage caused by plastic packing in the natural environment; most of them highlighted the necessity of plastic elimination inside the university, mainly in bars, vending machines, and canteens.

The word "Reduce" appeared 83 times, followed by "Meat" (74 times), most of the time, side by side. The second section focused on understanding the patterns of the diets. According to the literature [94], it was found that most of the Sapienza community is trying to decrease their consumption of red meat, and, at the same time, most of them followed a plant-based diet instead of an animal-based diet. Moreover, this work has already demonstrated the environmental impacts meat consumption brings. Therefore, it seems that people are aware of the issue and are changing their consumption habits.

"Price" was repeated 34 times, always regarding the decrease in prices of food considered sustainable. Healthy food tends to be more expensive than industrialized food. This is one of the critical issues of sustainable development, requiring immediate action from local, regional, and national authorities aiming to limit sales of unhealthy and unsustainable foods. Public policies need to be designed to create incentives for local producers of vegetables, legumes, and fruits. In this way, they could cultivate their food cheaply and sell it for a better price to the consumers. On the other hand, those items considered unhealthy should be penalized, using a rise in price to obtain a reduction in consumption.

Lastly, "Information" was cited 44 times, which means that people were urging for knowledge about sustainability. Additionally, they required more details about sustainable stores, especially for buying their products. Sustainability also was a very common word; however, not many people knew its full meaning. Ironically, those who desired more information were frequenting the university or working in its environment.

The results obtained in this research pointed out the importance of managing knowledge with the main aim of reaching wisdom, and from this point improve the value of the co-creation process [95,96]. More in detail, marketing knowledge management defined as "a determining factor in the competitive and sustainable growth of companies and international success" (Scuotto et al., 2022, p. 587) [97,98] also impacts on the competitive advantage by affecting organizational performance and innovation capabilities [97,99].

Food consumption and consumer behavior of Gen Z in the university context provide specific suggestions to improve policies directed to customer satisfaction.

## 6. Conclusions

This study aimed to understand the food consumption pattern within the Sapienza University community by analyzing the answers of a questionnaire fit for this purpose.

Considering all the results of the survey and the SCI equals 0.67, the Sapienza community partially aligns with the SDGs. It is important to remember that the sample is potentially interested in sustainability, since they voluntarily answered the questionnaire, leading to a possible bias. Then, from the previous analysis, it is possible to say that the sample showed small attention in purchasing food from sustainable places, especially sustainable fish retailers. Additionally, according to their answers, the University food supply should be improved in many aspects, such as the reduction in sales of plastic bottles and the amplification of sustainable and healthy foods stock—mainly plant-based products.

Even though most of the sample had full or enough knowledge about sustainability, only a quarter of them affirmed that their food consumption was sustainable. From that, a possible gap between intention and action was identified. A solution to the problems highlighted is certainly represented by the improvement of the level of knowledge within the community. Therefore, strong and mass advertising campaigns are needed, clearly informing the community about the linkage between food and sustainability and why it is important.

The sample also demonstrated awareness of the importance of waste segregation. However, people still use plastic bottles more than they should—one of the most harmful materials for the environment. Therefore, Sapienza University should restrict the sales of plastic bottles inside bars, canteens, and vending machines since the great majority of plastic utilization comes from these places.

In summary, the Sapienza community is partially committed to sustainable food consumption, having good indicators regarding reducing meat; consciousness about the importance of seasonal, local, and km0 products, and separation of waste. On the other hand, the university must improve in other areas by providing a better supply of healthy and sustainable foods, restricting plastic, and spreading more information regarding sustainability and food access.

The case of Sapienza shows that the relationship between food and sustainability is still not widespread among well-educated populations interested in the topic. This fact raises a concern when we think of low-income communities with limited access to education. Furthermore, considering the supply of unsustainable food within a renowned

university, such as Sapienza, it is worrisome to think how other institutions with fewer resources deal with the issue.

Since our survey is a cross-sectional study, one of its main limitations is the impossibility of inferring causality between the variables analyzed. At the same time, it possibly counts with a recall bias for being a questionnaire answered by a given sample—people who study or work at the university.

The survey conducted had the purpose of carrying out a first evaluation able to provide some basic indications regarding the awareness within Sapienza of the relationship between sustainability and food. Based on the indications obtained at this stage, it is expected to give rise to additional and in-depth investigations aimed at providing a model of sustainable food consumption that can be replicated on a large scale.

From this research, both theoretical and practical implications can be discussed. Firstly, the theoretical areas of food consumption, consumer behavior, and sustainability are strongly detected among Gen Z in the context of higher education. Additionally, the theoretical aspect of marketing knowledge management emerges as an area that can strongly contribute to improving the value creation or co-creation (by involving multiple actors) process.

Secondly, practical implications are related to the potential influence that food consumption and consumer behavior among Gen Z can have on the policies of the Sapienza University for merchandise and food distribution and consumption, with the main aim for improving the quality-of-service provisions to satisfy the customers' expectations.

Therefore, there is a need for greater dissemination of the topic to the mass population in a simple and accessible way. Additionally, investing in a more sustainable food supply within educational centers is fundamental for educating the next generation. Ultimately, populations worldwide are expected to become more aware of the environmental implications of food consumption, potentially advocating for a more sustainable future.

This paper has some limitations regarding the questionnaire and the methodology used. The semi-structured questionnaire does not allow for a deeper investigation of certain aspects. The other limitation is the qualitative methodology, which does not allow the generalization of the results, as it is a single case study. For the generation of the results, there would be a need to develop research, create multiple case studies to be compared, and improve our knowledge of the phenomenon. As for future directions, with the creation of this single case study, it is possible to create new research that identifies homogeneous or different universities to create comparative studies.

**Author Contributions:** Conceptualization, F.Z.; methodology, F.Z.; validation, C.D.R.; formal analysis, L.A.d.R.R. and C.D.R.; investigation, F.Z.; data curation, L.A.d.R.R.; writing—original draft preparation, L.A.d.R.R.; writing—review & editing, C.D.R. All authors have read and agreed to the published version of the manuscript.

**Funding:** This research received no external funding.

**Institutional Review Board Statement:** Not applicable.

**Informed Consent Statement:** Not applicable.

**Data Availability Statement:** Not applicable.

**Acknowledgments:** We wish to thank the British Council for funding this publication.

**Conflicts of Interest:** The authors declare no conflict of interest.

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
