# Peer review of "Needs of Sustainable Food Consumption in the Pandemic Era: First Results of Case Study"

_sustainability, doi:10.3390/su14169852_

Round 1
Reviewer 1 Report
Dear Authors,
We have appreciated very much this paper, which is quite interesting; however, the research needs some major and important adjustment and refinement.
We think that the authors should pay more attention to the fluency of the literature review, the results analysis, and the related discussion. As concerns the literature review, more specifically, please take into careful consideration the following references, which may result very useful in corroborating the theoretical background of the study, with peculiar reference to the Consumer Behavior
Please see some references below:
- Ltifi, M., & Hichri, A. (2021). The effects of corporate governance on the customer’s recommendations: a study of the banking sector at the time of COVID-19. Journal of Knowledge Management.
- Magni, D., Chierici, R., Fait, M., & Lefebvre, K. (2021). A network model approach to enhance knowledge sharing for internationalization readiness of SMEs. International Marketing Review.
- Bodhi, R., Luqman, A., Hina, M., & Papa, A. (2022). Work-related social media use and employee-related outcomes: a moderated mediation model. International Journal of Emerging Markets.
Most of all, we do think that a complete revision as concerns the meaning of the paper is quite opportune, with the Authors that should better adjust the flow of the reasoning in the evolution of the study. Some passages, in fact, can be more opportunely finalized, above all for providing evidence about the research implications, at scientific and managerial level. More in general, it is necessary to highlight the impact of research on the concept marketing knowledge management: the sense of this impact is quite arguable from the content of the research, but major emphasis is recommendable. This seems to be true also for the research limitations and future directions.
Finally, a linguistic revision is required.
Reviewer 2 Report
It is of great importance to ponder the SFS scientifically from the demanding side after a great deal from the supply side, this study provides a good example even though it is just from a university community's views. My suggestions are as follows, firstly, the sub-topic could be more distinct and specific, rather than using the obscure sentence " first results of case study"; secondly, the English grammar and words could be improved; and thirdly, the references could be filtrated.
Reviewer 3 Report
The manuscript studied the food consumption patterns of Gen Z students in the Sapienza University against sustainability, utilising the survey methodology. This is an interesting piece of work with potential however, the following major areas are advised to be addressed prior to consideration for publication:
The research motivation is clear. However, the research gap(s) being addressed by the manuscript is required to be articulated more clearly, along with specific, critical evaluation of similar works in the Introduction section.
The research questions, in their present form, have no linkage to the COVID-19 era and its effects on food consumption. So there is a misalignment in terms of what the research claims to have done against its formulated research questions.
There are serious grammatical and language flaws present in the current version of the manuscript and thus, a major overhaul towards fixing the language of the manuscript is highly needed.
Why did the researchers only focussed their data collection on 'Sapienza University'? This needs to be carefully justified in tha Methods and data section, and acknowledged openly as a limitation in the Conclusion section.
It is advised that key demographics of the sample studied are provided if possible, such as the age, gender, nationality (and any other similar, relevant characteristics). The analyses of these parameters may reveal commonalities/differences and carry the potential to further enrich the study.
The authors are advised to carry out a review of their findings against the extant literature, anchoring their findings and the implications of their research to the similar works in the current body of knowledge, in the Discussion section.
Round 2
Reviewer 1 Report
Congratulations I really appreciated the efforts in strengthen the paper
Reviewer 3 Report
The authors have now significantly improved their paper in line with the reviewer comments.